# Classification aware neural topic model for COVID-19 disinformation categorisation

**Xingyi Song**[1]*, **Johann Petrak**[1,2], **Ye Jiang**[1], **Iknoor Singh**[1,3], **Diana Maynard**[1], **Kalina Bontcheva**[1]

**1** Department of Computer Science, University of Sheffield, Sheffield, United Kingdom, **2** Austrian Research Institute for Artificial Intelligence, Vienna, Austria, **3** Panjab University, Chandigarh, India

* x.song@sheffield.ac.uk

**Data Availability Statement:** The full dataset is publicly available at: www.kaggle.com/dataset/fd97cd3b8f9b10c1600fd7bbb843a5c70d4c934ed83e74085c50b78d3db18443 The source code is

## Abstract

The explosion of disinformation accompanying the COVID-19 pandemic has overloaded fact-checkers and media worldwide, and brought a new major challenge to government responses worldwide. Not only is disinformation creating confusion about medical science amongst citizens, but it is also amplifying distrust in policy makers and governments. To help tackle this, we developed computational methods to categorise COVID-19 disinformation. The COVID-19 disinformation categories could be used for a) focusing fact-checking efforts on the most damaging kinds of COVID-19 disinformation; b) guiding policy makers who are trying to deliver effective public health messages and counter effectively COVID-19 disinformation. This paper presents: 1) a corpus containing what is currently the largest available set of manually annotated COVID-19 disinformation categories; 2) a classification-aware neural topic model (CANTM) designed for COVID-19 disinformation category classification and topic discovery; 3) an extensive analysis of COVID-19 disinformation categories with respect to time, volume, false type, media type and origin source.

## 1 Introduction

COVID-19 is not just a global pandemic, but has also led to an 'infodemic' ("an over-abundance of information") [1] and a 'disinfodemic' ("the disinformation swirling amidst the COVID-19 pandemic") [2]. The increased volume [3] of COVID-19 related disinformation has already caused significant damage to society; examples include: 1) **false treatments endangering health**, including disinformation [4] claiming that drinking alcohol can cure or prevent the new coronavirus, resulting in the deaths of more than 700 people from drinking denatured alcohol [5]; 2) **public mistrust**, including doctors being attacked because disinformation in WhatsApp claimed "health workers were forcibly taking away Muslims and injecting them with the coronavirus" [6]; 3) **public property damage**, including the burning of 5G masts caused by disinformation claiming they cause COVID-19 [7].

The ability to monitor and track at scale the categories of COVID-19 disinformation and the trends in their spread over time is an essential part of effective disinformation responses by media and governments. For instance, First Draft needed our COVID-19 disinformation

publicly available at: https://github.com/GateNLP/CANTM.

**Funding:** This research has been supported by European Union under grant agreement No.825297 WeVerify (https://weverify.eu/) and No. 825091 RISIS (https://www.risis2.eu/). There was no additional external funding received for this study. The funders had no role in study design, data collection and analysis, decision to publish, or preparation of the manuscript.

**Competing interests:** The authors have declared that no competing interests exist.

classifier to identify "data deficits" and track changing demand and supply of credible information on COVID-19 [8].

To enable such large-scale continuous monitoring and analysis, this paper presents a novel automatic COVID-19 disinformation classifier. It also provides an initial statistical analysis of COVID-19 disinformation in Section 5. The classifier is available both for research replicability and use by professionals (including those at the Agence France Presse (AFP) news agency and First Draft) The challenges of COVID-19 disinformation categorisation are that:

1. there is no sufficiently large existing dataset annotated with COVID-19 disinformation categories, which can be used to train and test machine learning models;

2. due to the time-consuming nature of manual fact-checking and disinformation categorisation, manual corpus annotation is expensive and slow to create. Therefore the classifier should train robustly from a small number of examples.

3. COVID-19 disinformation evolves quickly alongside the pandemic and our scientific understanding. Thus the model should provide suggestions about newly emerging relevant categories or sub-categories.

4. the classifier decisions should be self-explanatory, enabling journalists to understand the rationale for the auto-assigned category.

To address the first challenge, we created a new COVID-19 disinformation classification dataset. It contains COVID-19 disinformation debunked by the IFCN-led CoronaVirusFacts Alliance, and has been manually annotated with the categories identified in the most recent social science research on COVID-19 disinformation [3]. COVID-19 disinformation refers to false or misleading information related to COVID-19 that has potentially negative impacts. In this study, false claims debunked by the independent fact-checking members of the International Fact-Checking Network (IFCN) are deemed to be COVID-19 disinformation; no further selection criteria were applied.

To address the remaining three challenges, we propose a Classification-Aware Neural Topic Model (CANTM) which combines the benefits of BERT [9] with a Variational Autoencoder (VAE) [10, 11] based document model [12]. The CANTM model offers:

1. Robust classification performance especially on a small training set—instead of training the classifier directly on the original feature representation, the classifier is trained based on generated latent variables from the VAE [13]. In this case the classifier has never seen the 'real' training data during the training, thus reducing the chance of over-fitting. Our experiments show that combining BERT with the VAE framework improves classification results on small datasets, and is also scalable to larger datasets.

2. Ability to discover the hidden topics related to the pre-defined classes—the success of the VAE as a topic model (Some researchers distinguish 'document model' from 'topic model' [14, 15]. For simplicity, we consider both as a topic model.) has already been established in previous research [12, 14, 16]. We further adapt the VAE-based topic modelling to be classification-aware, by proposing a stacked VAE and introducing classification information directly in the latent topic generation.

3. The classifier is self-explaining—in CANTM the same latent variable (topic) is used both in the classifier and for topic modelling. Thus the topic can be regarded as an explanation of the classification model. We further introduce 'class-associated topics' that directly map the topic words to classifier classes. This enables the inspection of topics related to a class, thus

providing a 'global' explanation of the classifier. In addition, BERT attention weights could also be used to explain classifier decision, but this is outside the scope of this paper.

Our experiments in Section 4 compare CANTM classification and topic modelling performance against several state-of-the-art baseline models, including BERT and the Scholar supervised topic model [16]. The experiments demonstrate that the newly proposed CANTM model has better classification and topic modelling performance (in accuracy, average F1 measure, and perplexity) and is also more robust (measured in standard deviation) than the baseline models.

The main contributions of this paper are:

1. A new COVID-19 disinformation corpus with manually annotated categories.

2. A BERT language model with an asymmetric VAE topic modelling framework, which shows performance improvement (over using BERT alone) in a low-resource classifier training setting.

3. The CANTM model, which takes classification information into account for topic generation.

4. The use of topic modelling to introduce 'class-associated' topics as a global explanation of the classifier.

5. An extensive COVID-19 disinformation category analysis.

6. The corpus and source code of this work are open-source, and the web service and API are publicly available (please refer to Section 9 for details).

## 2 Dataset and annotation

The dataset categorises according to topic false claims about COVID-19, which were debunked and published on the IFCN Poynter website (https://www.poynter.org/ifcn-covid-19-misinformation/). The dataset covers debunks of COVID-19-related disinformation from over 70 countries and 43 languages, published in various sources (including social media platforms, TV, newspapers, radio, message applications, etc.).

The structure of the data is illustrated in Table 1 (for a full description of all label fields in the table, please refer to S1 Appendix). Each dataset entry includes 9 different fields. Fields **a** to **d** are extracted directly from HTML tags in the IFCN web page. Besides the manually-assigned category label (field **i**), we also apply various Natural Language Processing (NLP) tools to

**Table 1. COVID-19 disinformation category data structure.**

| Label Fields | Extraction Method | Example |
|---|---|---|
| a. Debunk Date | IFCN HTML | 2020/04/09 |
| b. Claim | IFCN HTML | A photograph . . . lockdown. |
| c. Explanation | IFCN HTML | The photo was . . . officer. |
| d. Source link | IFCN HTML | factcheck.afp.com/photo-was. . . |
| e. Veracity | String Match | False |
| f. Originating platform | String Match | Facebook, Twitter, Instagram |
| g. Source page language | langdetect | English |
| h. Media Types | JAPE Rule | Image |
| i. **Categories** | Manually annotated | Prominent actors |

automatically extract and refine the information contained in fields **e** (Veracity), **f** (Claim Origin), **g** (Source page language), **h** (Media Types).

The manual labelling of the dataset entries into disinformation categories was conducted as part of the EUvsVirus hackathon (https://www.euvsvirus.org/). We defined 10 different COVID-19 disinformation categories based on [3]: (i) Public authority; (ii) Community spread and impact; (iii) Medical advice, self-treatments, and virus effects; (iv) Prominent actors; (v) Conspiracies; (vi) Virus transmission; (vii) Virus origins and properties; (viii) Public Reaction; (ix) Vaccines, medical treatments, and tests; and (x) Other. Please refer to S4 Appendix for the full description of these categories.

During the hackathon 27 volunteer annotators were recruited amongst the hackathon participants. The annotation process undertaken as part of the WeVerify project has received ethical clearance from the University of Sheffield Ethics Board. The volunteer annotators who manually categorised the COVID-19 false claims were provided with the project's information sheet alongside the instructions for data annotation. As all annotations were carried out via an online data annotation tool, consent was obtained verbally during the virtual annotator information sharing and training session. The dataset contains false claims and IFCN debunks in English published until 13th April, 2020 (the hackathon end date). The claim, the fact-checkers' explanation and the source link to the fact-checkers' own web page were all provided to the annotators. The volunteers were trained to assign to each false claim the most relevant of the 10 COVID-19 disinformation categories and to indicate their confidence (on a scale of 0 to 9). The English claims were randomly split into batches of 20 entries. In the first round, all annotators worked on unique batches. In the second round, they received randomised claims from the first round, so inter-annotator agreement (IAA) could then be measured.

The volunteers annotated 2,192 false claims and their debunks (see Table 2). Amongst these, 424 samples were double- or multiple-annotated, from which we calculated the IAA. At this stage, vanilla Cohen's Kappa [17] was only 0.46.

To increase the data quality and provide a good training sample for our ML model, we applied a cleaning step to filter low quality annotations. We first measured annotator quality by observing agreement change when removing an (anonymous) annotator. This annotator quality was scored based on the magnitude of score variance. Based on this, the annotations from the two annotators with the lowest scores were removed.

We also measured the impact of annotator confidence score on annotation agreement and the amount of filtered data, and set a confidence threshold for each annotator, based on the quality check from the first round (for most annotators, this threshold was 6). Any annotation with confidence below this threshold was filtered out.

Ultimately, 1,293 debunks remained with at least one reliable classification, and IAA rose to 73.36% and Cohen's Kappa to 0.7040.

The final dataset was produced by merging the multiple-annotated false claims on the basis of: 1) majority agreement between the annotators where possible; 2) confidence score—if

**Table 2. Label counts and annotation agreements of unfiltered annotation (All) and filtered annotation (Cleaned).**

|  | All | Cleaned |
|---|---|---|
| Single Annotated | 1056 | 1038 |
| Double Annotated | 213 | 186 |
| Multiple Annotated | 211 | 69 |
| Annotation Agreement | 0.5145 | 0.7336 |
| Kappa | 0.4660 | 0.7040 |

**Table 3. Number of examples per category in the final dataset.**

| PubAuthAction | CommSpread | PubRec | PromActs |
|---|---|---|---|
| 251 | 225 | 60 | 221 |
| GenMedAdv | VirTrans | Vacc | Consp |
| 177 | 80 | 76 | 97 |
| VirOrgn | None | | |
| 63 | 43 | | |

there was no majority agreement, the label with the highest confidence score was adopted. Table 3 shows the statistics of the merged dataset for each of the ten categories. Category distribution is consistent with that found in [3].

## 3 Classification aware neural topic model

This section begins with a brief overview of related work on topic models, which is a necessary background motivation for our CANTM model, which is described in Section 3.1. Other related work is reviewed in Section 7.2.

Miao et. al. [12] introduce a generative neural variational document model (NVDM) that models the document ($x$) likelihood $p(x)$ using a variational autoencoder (VAE), which can be described as:

$$
\begin{aligned}
\log p(x) &= ELBO + D_{KL}(q(z|x)||p(z|x)) \\
ELBO &= \mathbb{E}_{q(z|x)}[\log p(x|z)] - D_{KL}(q(z|x)||p(z))
\end{aligned}
\tag{1}
$$

Where $p(z)$ is the prior distribution of latent variable $z$, $q(z|x)$ is the inference network (encoder) used to approximate the posterior distributions $p(z|x)$ and $p(x|z)$ is the generation network (decoder) to reconstruct the document based on latent variable (topics) $z \sim q(z|x)$ sampled from the inference network.

According to Eq 1, maximising the ELBO (evidence lower bound) is equivalent to maximising the $p(x)$ and minimising the Kullback–Leibler divergence ($D_{KL}$) between $q(z|x)$ and $p(z|x)$. Therefore, maximising ELBO will be the objective function in the NVDM or VAE framework, or negative ELBO for gradient descent optimisation. The latent variable $z$ then can be treated as the latent topics of the document.

NVDM is an unsupervised model, hence we have no control on the topic generation. In order to uncover the topics related to the target $y$ (e.g. category, sentiment or coherence) in which we are interested, we can consider several previous approaches. The Topic Coherence Regularization (NTR) [18] applies topic coherence as additional loss (i.e. loss $\mathcal{L} = -ELBO + C$) to regularise the model and generate more coherent topics. SCHOLAR [16] directly inserts the target information into the encoder (i.e. $q(z|x, y)$), making the latent variable also dependent on the target. However, when target information is missing at application time, SCHOLAR treats the target input as a missing feature (i.e. all zero vector) or all possible combinations. Hence the latent variable becomes less dependent on the target.

Inspired by the stacked VAE of [13], we combined ideas from NTR and SCHOLAR. In particular, we stacked a classifier-regularised VAE (M1) and a classifier-aware VAE (M2) enabling the provision of robust latent topic information even at testing time without label information.

## 3.1 Model detail

The training sample $D = (x, x_{bow}, y)$ is a triple of the BERT word-pieces sequence representation of the document ($x$), a bag-of-words representation of the document ($x_{bow}$) and its associate target label $y$.

The general architecture of our model is illustrated in Fig 1. CANTM is a stacked VAE containing 6 sub-modules:

1. M1 encoder (or M1 inference network) $q(z|x)$

2. M1 decoder (or M1 generation network) $p(x_{bow}|z)$

3. M1 Classifier $\hat{y} = f(z)$

4. M1 Classifier decoder $p(x|\hat{y})$

5. M2 encoder (or M2 inference network) $q(z_s|x, \hat{y})$

6. M2 decoder (or M2 generation network) $p(x_{bow}|\hat{y}, z_s)$ and $p(\hat{y}|z_s)$

Sub-modules 1 and 2 implement a VAE similar to NVDM. The modification over the original NVDM is that instead of bag-of-words ($x_{bow}$) input and output to the model, our input is a BERT word-pieces sequence representation of the original document ($x$). The reason for this modification is that $x$ can be seen as a grammar-enriched $x_{bow}$, and we could capture better semantic representation in the hidden layers (e.g. though pre-trained BERT) and thus benefit the classification and topic generation. Also, $q(z|x)$ is an approximation of $p(z|x_{bow})$, and they do not have to follow the same condition [10], as our model is still under the VAE framework. Sub-modules 5 and 6 implement another VAE that models the joint probability of document $x_{bow}$ and label $\hat{y}$. Note that the label in M2 is a classifier prediction, hence this label information will always be available for M2 VAE. To apply CANTM to unlabelled test data, we fix the M1

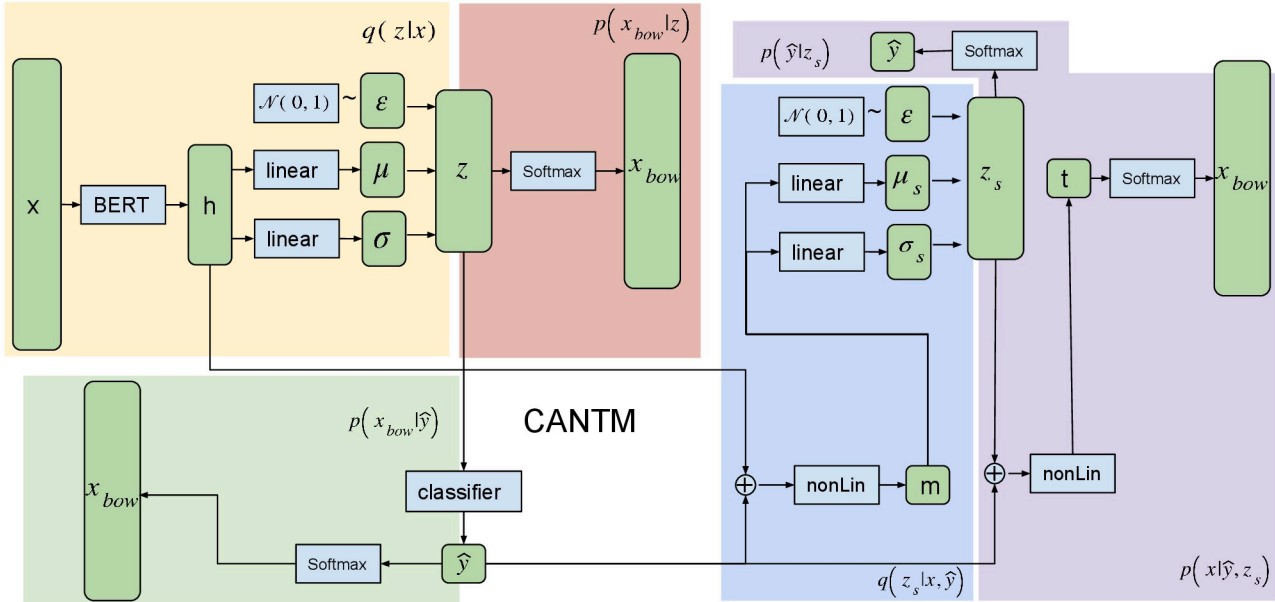

**Fig 1. Overview of model architecture, linear block is the linear transformation (i.e. linear(x) = Wx + b), nonLin is linear transformation with non-linear activation function f(linear(.)), softmax is softmax activated linear function.**

weights that are pre-trained on the labelled data, and only train the M2 model. In Sections 3.1.1 to 3.1.5, we will describe each sub-module in detail.

**3.1.1 M1 encoder.** The M1 encoder is illustrated in the yellow part of Fig 1. During the encoding process, the input $x$ is first transformed into a BERT-enriched representation $h$ using a pre-trained BERT model. We use the *CLS* token output from BERT as $h$. Then linear transformations $linear_1(h)$ and $linear_2(h)$ transform the $h$ into parameters of variational distribution that are used to sample the latent variable $z$.

$$linear_k(h) = W_k h + b_k \qquad (2)$$

Where $W_k$ and $b_k$ are weight and bias vectors respectively for linear transformations $k$.

The variational distribution is a Gaussian distribution ($\mathcal{N}(\mu, \sigma)$) The M1 Encoder is represented in Eq 3.

$$q(z|x) = \mathcal{N}(\mu, \sigma)$$
$$\mu = linear_1(h), \sigma = linear_2(h) \qquad (3)$$
$$h = BERT(x)$$

Following previous approaches [10–12], a re-parameterisation trick is applied to allow back-propagation to go though the random node.

$$z = \mu + \sigma \odot \epsilon, \epsilon \sim \mathcal{N}(0, 1) \qquad (4)$$

where $\epsilon$ is random noise sampled from a 0 mean and variance 1 Gaussian distribution. In the decoding process (described next), the document is reconstructed from the latent variable $z$, hence $z$ can be considered as the document topic.

**3.1.2 M1 decoder.** The decoding process (the red part in Fig 1) reconstructs $x_{bow}$ from the latent variable $z$. This is modelled by a fully connected feed-forward (FC) layer with softmax activation (sigmoid activation normalised by softmax function. For the rest of the paper we will describe this as softmax activation for simplicity). The likelihood of the reconstruction $p(x_{bow}|z)$ can be calculated by

$$p(x_{bow}|z) = softmax(zR + b) \odot x_{bow}$$

Where $R \in \mathbb{R}^{|z| \times |V|}$, and $|V|$ is the vocabulary size. $R$ is a learnable weight for mapping between topics and words. The topic words for each topic can be extracted according to this weight. $\odot$ is the dot product.

**3.1.3 M1 classifier and classifier decoder.** The classifier $\hat{y} = softmax(FC(z))$ is a softmax activated FC layer. It is based on the same latent variable $z$ as the M1 encoder. Since the M1 VAE and classifier are jointly trained based on $z$, it can be seen as a 'class regularized topic' and also serve as a 'global explanation' of the classifier. Furthermore, $\hat{y}$ itself can be seen as a compressed topic of $z$, or 'class-associated topic'. The document can be reconstructed by $\hat{y}$ in the same way as the M1 decoder, and the likelihood of $p(x_{bow}|\hat{y})$ is given by:

$$p(x_{bow}|\hat{y}) = softmax(\hat{y}R_{ct} + b) \odot x_{bow}$$

where $R_{ct} \in \mathbb{R}^{|y| \times |V|}$ is a learnable weight for 'class-associated topic' word mapping.

**3.1.4 M2 encoder.** The encoding process of M2 (the blue part in Fig 1) is similar to M1, but instead of only encoding $x$, M2 encodes both the document and the predicted label from the M1 classifier $q(z_s|x, \hat{y})$. In the M2 encoder process, we first concatenate ($\oplus$) the BERT representation $h$ and predicted label $\hat{y}$, then merge them through a leaky rectifier (*LRelu*) [19]

activated FC layer. We refer to this as $nonLin_n$ in the remainder of the paper.

$$m \quad = nonLin_1(h \oplus \hat{y})$$
$$= LRelu(FC(h \oplus \hat{y}))$$

As for the M1 encoder, a linear transformation then maps the merged feature $m$ to the parameters of the variational distribution represented by the latent variable of M2 model $z_s$. The variational distribution is a Gaussian $\mathcal{N}(\mu_s, \sigma_s)$:

$$q(z_s|x, \hat{y}) = \mathcal{N}(\mu_s, \sigma_s)$$
$$\mu_s = linear_3(m), \sigma_s = linear_4(m)$$

**3.1.5 M2 decoder.**   The decoding process of M2 $p(x_{bow}, \hat{y}|z_s)$ is divided into two decoding steps $(p(x_{bow}|\hat{y}, z_s)$ and $p(\hat{y}|z_s))$ by Bayes Chain Rule.

The step $p(\hat{y}|z_s)$ can be considered as M2 classifier, calculated by softmax FC layer, the likelihood function is modelled as $p(\hat{y}|z_s) = softmax(FC(z_s)) \odot \hat{y}$. The M2 classifier will not be used for classification in this work, only for the loss calculation (see Section 3.1.6).

In step $p(x_{bow}|\hat{y}, z_s)$, we first merge $\hat{y}$ and $z_s$ using $nonLin$ layer

$$t = nonLin_2(\hat{y} \oplus z_s)$$

Where $t$ is a 'classification aware topic'. Then $x_{bow}$ is reconstructed using a softmax layer. The likelihood function is:

$$p(x|\hat{y}, z_s) \quad = softmax(tR_s + b) \odot x_{bow}$$

where $R_s \in \mathbb{R}^{|z_s| \times |V|}$ is a learnable weight for the 'classification aware topic' word mapping.

**3.1.6 Loss function.**   The objective of CANTM is to: 1) maximise $ELBO_{x_{bow}}$ for M1 VAE; 2) maximise $ELBO_{x_{bow}, \hat{y}}$ for M2 VAE; 3) minimise cross-entropy loss $\mathcal{L}_{cls}$ for M1 classifier and 4) maximise the log likelihood of M1 class decoder $\log[p(x_{bow}|\hat{y})]$. Hence the loss function for CANTM is

$$\mathcal{L} \quad = \lambda \mathcal{L}_{cls} - ELBO_{x_{bow}} - ELBO_{x_{bow}, \hat{y}}$$
$$- \mathbb{E}_{\hat{y}}[\log p(x_{bow}|\hat{y})]$$
$$= \lambda \mathcal{L}_{cls} - \mathbb{E}_z[\log p(x_{bow}|z)] + D_{KL}(q(z|x)||p(z))$$
$$- \mathbb{E}_{z_s}[\log p(x_{bow}|\hat{y}, z_s)] - \mathbb{E}_{z_s}[\log p(\hat{y}|z_s)]$$
$$+ D_{KL}(q(z_s|x, \hat{y})||p(z_s)) - \mathbb{E}_{\hat{y}}[\log p(x_{bow}|\hat{y})]$$

where $p(z)$ and $p(z_s)$ are zero mean diagonal multivariate Gaussian priors ($\mathcal{N}(0, I)$), $\lambda = vocabSize/numclass$ is a hyperparameter controlling the importance classifier loss. For full details of the ELBO term deriving process please see S5 Appendix).

## 4 CANTM experiments

In this section, we compare the classification and topic modelling performance of CANTM against state-of-the-art baselines (BERT [9], SCHOLAR [16], NVDM [12], and LDA [20]), as well as human annotators.

The details of experiment settings for each model are described below:

- BERT [9]: We use Huggingface [21] 'BERT-based-uncased' pre-trained model and the Pytorch implementation in this experiment. As with CANTM, we use BERT [CLS] output as

BERT representation, and an additional 50 dimensional feed-forward hidden layer (with leaky ReLU activation) after that. CANTM contains a sampling layer after the BERT representation, this additional layer is added for fair comparison. Please check S5 Appendix on impact of the additional hidden layer. Only the last transformer encoding layer (layer 11) is unlocked for fine-tuning, the rest of the BERT weights were frozen for this experiment. The Pytorch (https://pytorch.org/) implementation of the Adam optimiser [22] is used in the training with default settings. The batch size for training is 32. All BERT-related (CANTM, NVDMb) implementations in this paper follow the same settings.

- CANTM (our proposed method): We use the same BERT implementation and settings as described above. The sampling size (number of samples $z$ and $z_s$ drawn from the encoder) in training and testing are 10 and 1 respectively, and we only use expected value ($\mu$) of $q(z|x)$ for the classification at testing time. Unless mentioned otherwise, the topics reported from CANTM are 'classification-aware'.

- NVDM [12]: We re-implement NVDM Based on code at https://github.com/YongfeiYan/Neural-Document-Modeling, with two versions: 1) original NVDM as described in [12] ("NVDMo" in the results); 2) NVDM with BERT representation ("NVDMb" in the results).

- SCHOLAR [16]: We use the original author implementation from https://github.com/dallascard/scholar with all default settings (except the vocabulary size and number of topics).

- Latent Dirichlet Allocation (LDA) [20]: the Gensim [23] implementation is used.

The input for each disinformation instance is the combination of the text of the false Claim and the fact-checkers' Explanation (average text length 23 words), while the vocabulary size for topic modelling is 2,000 words (S6 Appendix—Experimental Details provides additional detail on the parameters setting).

Table 4 shows average accuracy (Acc), macro F-1 measure (F-1). The F-1 is calculated as the average F-1 measure of all classes. and perplexity (Perp.), based on 5-fold cross-validation. Standard deviation is reported in parentheses. The majority class is 'Public authority action ('PubAuth') at 19.4%).

To ensure fair comparison between CANTM and the BERT classifier, we first compared: 1) BERT with an additional hidden layer that matches the dimension of latent variables (denoted BERT in the result); 2) BERT without the additional hidden layer, i.e. applying BERT [CLS] token output directly for classification (denoted BERTraw in the Table 4). According to our results, BERT with the additional hidden layer has better performance in both accuracy and F-measure. Therefore, unless mentioned otherwise thereon 'BERT' refers to BERT with the additional hidden layer.

**Table 4. Five-fold cross-valuation classification and topic modelling results, n/a stands for not applicable for the model.** The standard deviation is shown in parentheses. The majority class is 'PubAuth' at 19.4%.

|  | Acc. | F-1 | Perp. |
|---|---|---|---|
| Bert | 58.78(3.36) | 54.19(6.85) | n/a |
| BERTraw | 58.77(3.56) | 49.74 (7.62) | n/a |
| Scholar | 48.17(6.78) | 36.40(10.85) | 2947(353) |
| NVDMb | n/a | n/a | 1084(88) |
| NVDMo | n/a | n/a | 781(35) |
| LDA | n/a | n/a | 8518(1132) |
| CANTM | **63.34(1.43)** | **55.48(6.32)** | **749(63)** |

**Table 5. COVID-19 disinformation class level F1 score, standard deviation in parentheses.**

|          | PubAuth        | CommSpread      | MedAdv         | PromActs       | Consp          |
|----------|----------------|-----------------|----------------|----------------|----------------|
| BERT     | 61.17(4.50)    | 62.27(5.83)     | 75.03(6.54)    | 60.12(3.25)    | 49.92(12.04)   |
| BERTraw  | **65.64(2.91)**| 59.35(4.77)     | 75.82(5.53)    | 65.51(4.34)    | 41.90 (10.46)  |
| SCHOLAR  | 47.92(9.77)    | 48.84(11.56)    | 71.11(6.99)    | 46.93(8.66)    | 31.30(13.78)   |
| CANTM    | 64.35(1.44)    | **66.50(3.87)** | **79.68(2.12)**| **67.21(3.72)**| **60.06(6.80)**|
|          | VirTrans       | VirOrgn         | PubRec         | Vacc           | None           |
| BERT     | **42.67(8.70)**| **57.62(6.72)** | 23.68(10.01)   | 64.62(9.66)    | 12.59(11.35)   |
| BERTraw  | 41.42(5.36)    | 53.20(15.92)    | **27.19(13.55)**| 65.48(9.62)   | 1.90 (3.8)     |
| SCHOLAR  | 11.71(10.06)   | 45.15(20.49)    | 5.71(11.42)    | 55.37(15.78)   | 0.0(0.0)       |
| CANTM    | 40.21(8.56)    | 55.19(3.43)     | 25.04(9.87)    | **72.28(8.40)**| **15.52 (15.0)**|

BERT as a strong baseline outperforms SCHOLAR in accuracy by more than 10%, and almost 18% F-1 measure. This is expected, because BERT is a discriminative model pre-trained on large corpora and has a much more complex model structure than SCHOLAR.

Our CANTM model shows an almost 5% increase in accuracy and more than 1% F-1 improvement over BERT. Note that CANTM not only improves the accuracy and F1 measure over the best performing BERT baseline, but it also improves standard deviation. Training on latent variables with multi-task loss is thus an efficient way to train on a small dataset even with a pre-trained embedding/language model. In the topic modelling task, CANTM has the best (lowest) perplexity performance compared with the traditional unsupervised topic model LDA, VAE based unsupervised topic model NVDM variants (NVDMo and NVDMb) and the supervised neural topic model Scholar.

Table 5 shows the class-level F1 score on the COVID-19 disinformation corpus. CANTM has the best F1 score over most of the classes (CommSpread, MedAdv, PromActs, Consp, Vacc, None), also with better standard deviations. Except for the None class, standard deviations for CANTM are below 10. From the results, the most difficult class to assign is 'None'. It represents disinformation that the annotators struggled to classify into one of the other 9 categories and is therefore topically very broad.

The human vs CANTM classification comparison is shown in Fig 2. Fig 2a is a percentage stacked column chart of CANTM category prediction based on 5-fold cross-validation (please refer to S7 Appendix for the confusion matrix). Each column represents the percentage of the predicted category (in a different colour) by CANTM. For example, amongst all disinformation manually labelled as 'Public authority action' (the 'PubAuth Column'), 69.3% is correctly labelled by CANTM (shown in blue) and 12.4% is incorrectly labelled as 'Prominent actors' (shown in dark green).

Fig 2b is a percentage stacked column chart of human agreements according to pairwise agreement. The colour in each column represents the percentage of annotator agreement/disagreement in a given category. Our annotation agreement was measured pairwise, therefore each column represents all disinformation that was annotated in a certain category by at least one annotator, and the colours in each column represent the percentage of the category annotated by another annotator. For example, for all disinformation annotated as Public authority action by at least one annotator (the 'PubAuth Column') 60.2% of the time another annotator also annotated it as Public authority action (shown in blue). This also means that the agreement percentage for the Public authority action class is 60.2%. The annotators disagreed on the remaining 39.8%, with 12.4% of them the second annotator annotated the instance as 'Prominent actors' (shown in dark green), and 6.2% of the time as 'Community spread'(red colour).

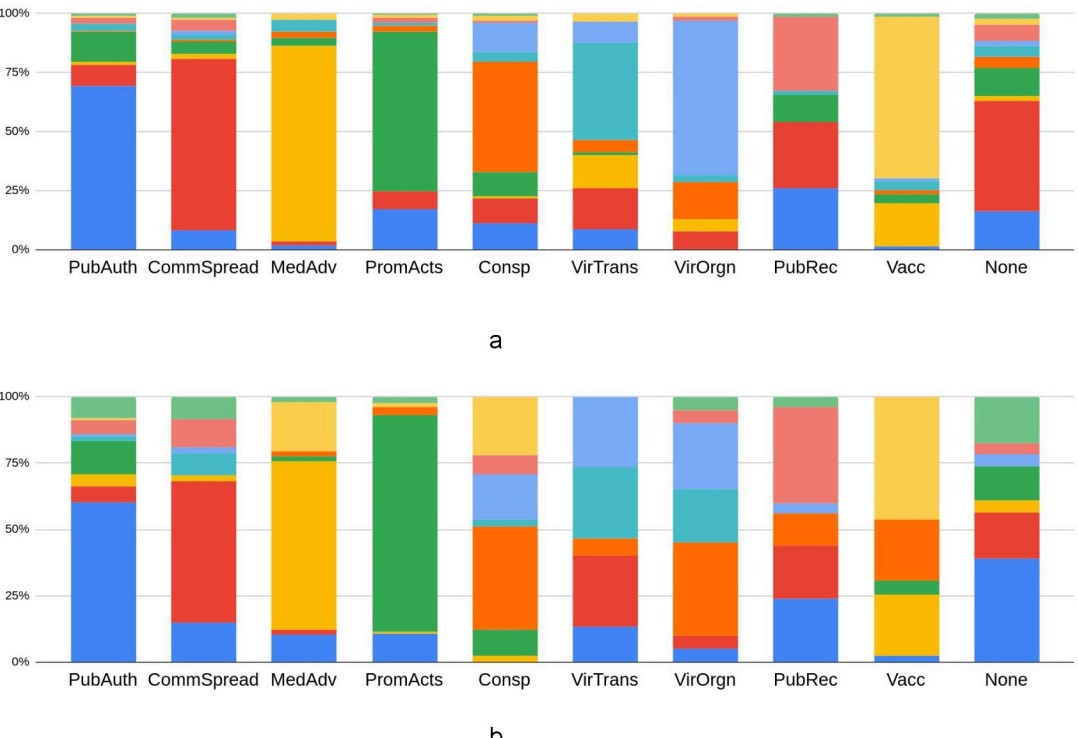

**Fig 2.** a. Percentage stacked column chart of CANTM category prediction b. Percentage stacked column chart of human agreements in the pairwise agreement measurement.

By comparing Fig 2a and 2b, we can see that the percentages of CANTM errors and human disagreement generally follow a similar distribution. The three categories where CANTM has the lowest accuracy/ recall (Other:2.3%, Public preparedness: 31.3% and Virus Transmission: 41.3%) are also the three categories with the lowest agreement between the human anotators (None: 8.3%, Public preparedness: 41.3% and Virus Transmission: 47.1%).

CANTM prediction performance also depends on the number of instances available for training (Table 3 shows the number of manual labels in each category available for training). The categories 'Public authority action', 'Community spread', 'Prominent actors' and 'General medical advice' have a relatively high number of instances ($> = 177$ instances) and also have better classification performance than other classes. In addition, according to Fig 2, 'General medical advice' and 'Vaccine development' have high disagreement between annotators. Classification error, however, is higher for the 'Vaccine development' category. This may be because the number of training instances for the 'General medical advice' category is almost triple that of 'Vaccine development'; thus the model is more biased towards the former.

In general, the overall CANTM performance (accuracy: 63.34%, or agreement between CANTM and the human annotators) is better than human inter-annotator agreement prior to the filtering/cleaning process (51.45%).

## 5 COVID-19 disinformation analysis and discussion

As discussed above, the creation of the CANTM classifier was motivated by the journalists' and fact-checkers' needs for in-depth, topical analysis and monitoring of COVID-19 disinformation. Therefore, we also conducted a statistical analysis of debunked COVID-19

**Table 6. Statistics of debunked COVID-19 disinformation by IFCN members.** (1 January—30 June 2020).

| Category | PubAuth | CommSpread | PubRec | PromActs | MedAdv |
|---|---|---|---|---|---|
| | 1672 | 1527 | 301 | 1160 | 1115 |
| | VirTrans | Vacc | Consp | VirOrgn | Other |
| | 330 | 396 | 809 | 151 | 148 |
| Media Type | Video | Text | Audio | Image | Not Clear |
| | 1774 | 3317 | 144 | 1647 | 897 |
| Veracity | False | Part. False | Misleading | No Evid. | Other |
| | 6392 | 330 | 733 | 94 | 63 |
| Platform | Twitter | Facebook | WhatsApp | News | Blog |
| | 1198 | 4333 | 1023 | 464 | 91 |
| | LINE | Instagram | Oth. Social | Oth. msg | TV |
| | 83 | 94 | 542 | 44 | 21 |
| | TikTok | YouTube | Other | | |
| | 17 | 279 | 949 | - | - |
| Country | Spain | India | Brazil | US | Other |
| | 484 | 1503 | 471 | 872 | 4282 |
| Language | EN | ES | PT | FR | Other |
| | 2880 | 1385 | 540 | 421 | 2386 |

disinformation during the first six months of 2020, with respect to its category, the type of media employed, the social media platform where it originated, and the claim veracity (e.g. false, misleading).

7609 debunks of COVID-19 disinformation were published by IFCN members between 1st January and 30th June 2020 and were the focus of our study here. Each false claim was categorised by our trained CANTM model into one of the ten topical categories. Table 6 shows that the two most prevailing categories were disinformation about government and public authority actions (PubAuth) and the spread of the disease (CommSpread), which is consistent with the findings of the earlier small-scale social science study by [3]

With respect to platform of origin, as shown in Table 6, Facebook was was leading source with more than 45% of disinformation published there. Moreover, 3.6 times more false claims originated on Facebook as compared to the second highest source, Twitter. Unfortunately, the majority research into disinformation has focused on Twitter [24–34] rather than Facebook, due to the highly restricted data access and terms and conditions of the latter.

To capture the longitudinal changes, we calculated weekly trends of the number of debunked disinformation (see Fig 3). The solid light green line represents the the weekly number of debunked disinformation while the dashed orange line is the number of worldwide Google searches for 'Coronavirus' (https://trends.google.com/trends/explore?q=%2Fm%2F01cpyy). Debunked disinformation was normalised to make it comparable to the Google search trends. We used the same normalisation method as Google search, i.e. the percentage of debunked disinformation compared to the week with the highest number of debunked disinformation (week 29/03/2020 with 810 debunks). The highest normalised value is thus 100 in both cases.

The number of Google searches reflects global public interest in COVID-19. As shown in Fig 3, the trends in debunked disinformation over time are similar to those for Google searches, with a slight temporal delay which is likely due to the time required for fact-checking.

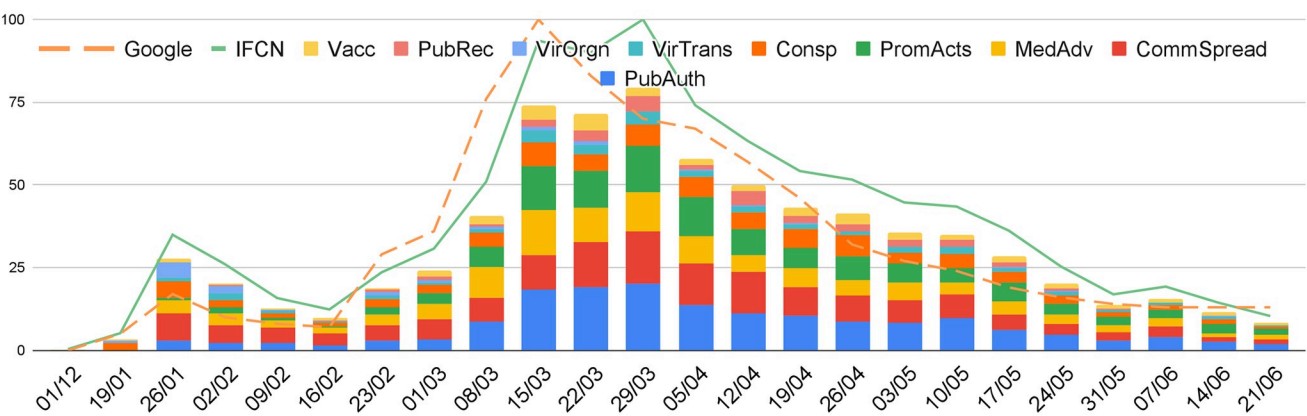

**Fig 3. Weekly trends of normalised IFCN debunks, COVID related Google searches and categories.**

The two trends also demonstrate that disinformation volume is proportional to the information need of the general population. Both numbers start to grow from the middle of January, and reach 2 peaks in the January to June period: the smaller peak is at the end of January, and the second peak in the middle of March. It is likely that the two peaks are related to the WHO announcement of Public Health Emergency of International Concern on 30 January, 2020 and the COVID-19 pandemic on 11 March, 2020. Searches and disinformation both started to decay after the second peak.

The column chart on Fig 3) shows the proportion of each disinformation category (in a different colour) on a weekly basis. At the beginning, the most widespread disinformation category is 'Conspiracy theory'. Between the end of January and mid February the prevailing categories become 'Community spread' and 'Virus origin'. On February 9, WHO reported [35] that the number of COVID-19 deaths rose to 813 and exceeded the number of deaths during the SARS-CoV (severe acute respiratory syndrome coronavirus) outbreak. 'General medical advice' soon became the most highly spread disinformation category until early March. Soon after the pandemic announcement from WHO on March 11th, 'Public authority action' became the top disinformation category and remained thereafter. Other widespread categories after mid-March include 'Community Spread' and 'Prominent actors'. In contrast, disinformation about 'Virus Origin' became much less widespread after March.

We also investigated the question of the modalities employed by disinformation from the different topical categories. Fig 4 shows a percentage stacked column chart per category of the modality of the disinformation claims in this category, i.e. image, video, text, or audio. The modality information is extracted automatically using rule-based patterns applied to the 'Claim', 'Explanation', 'Claim Origin' and 'Source page' (though 'Source Link') of the published debunks. For details on the rule-based extractor see S3 Appendix. The last column (All) in the figure is the overall distribution of media types.

In general, Fig 4 shows that about half of the disinformation was spread through primarily textual narratives (e.g. text messages, blog articles). Video and image-based disinformation account for around a quarter of all media forms respectively, while only 2.1% of COVID-19 disinformation was spread by audio.

At the category level, although textual narratives are the predominant media for most categories ('Public authority action', 'General medical advise', 'Prominent actors', 'Conspiracy theories', 'Virus transmission' and 'Vaccine development'), around 50% of false claims about 'Virus origin' and 'Public Preparedness' are spread through video. Image-based disinformation

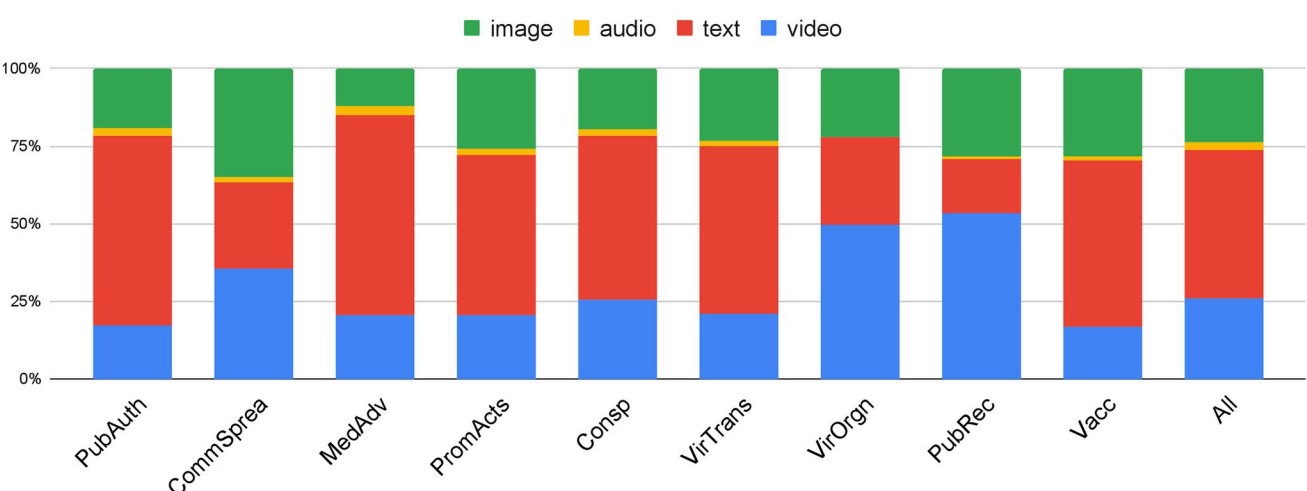

**Fig 4. Percentage stacked column chart of media type vs. category.**

is not dominant in any category, although along with video it has a relatively high percentage in disinformation about 'Community Spread'.

The third key research question was concerned with the role of social media platforms and messaging apps in the COVID-19 disinfodemic. Fig 5 is a percentage stacked column chart, which shows on a per social platform/app basis a breakdown of the categories of disinformation that circulated on that given platform/app. The originating platforms/apps considered in this study are shown in Table 6. The information about originating platform is extracted automatically from HTML tags in the IFCN web page of each debunk and is post-processed through string matching described in S2 Appendix.

As shown in Fig 5, the category distribution across different social media platforms (Facebook, Twitter, Instagram etc.) are similar, while the most widespread categories are 'Public Authority action' and 'Community Spread'. However, Instagram has a considerably larger percentage of disinformation in the 'virus origin' category—10.9% for Instagram compared against less than 2% on other social media platforms. This may be because Instagram has a higher proportion of video media than the other platforms, and according to our previous finding (Fig 4) 'Virus origin' is frequently spread through videos. The percentage of 'Virus origin' is also relatively high on the video platform YouTube (7.2%). 'Conspiracy theory'

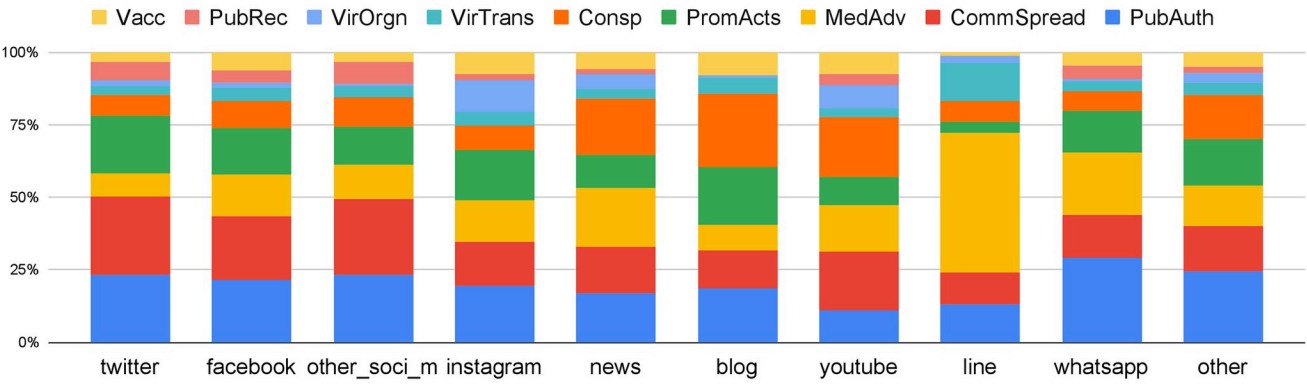

**Fig 5. Percentage stacked column chart of claim origin vs. category.**

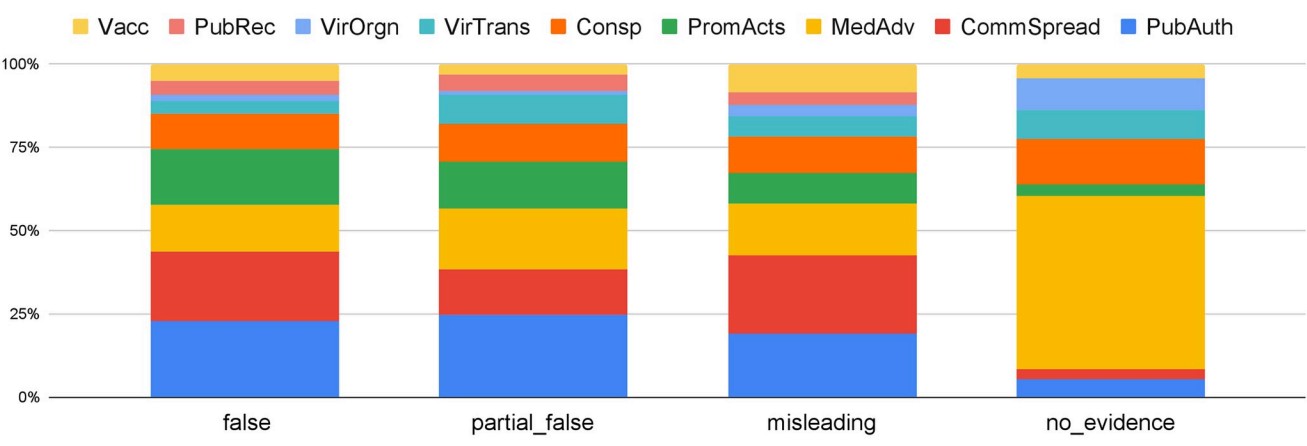

**Fig 6. Percentage stacked column chart of veracity type vs. category.**

disinformation is spread primarily through news, YouTube, and blog posts, than through other social media platforms and messaging apps (LINE and WhatsApp). This may be related to the lengthier nature of conspiracy theory narratives and videos, which are thus better suited to news, YouTube, and blog posts. In contrast, messaging apps (LINE and WhatsApp) have a much higher proportion of 'General medical advice' disinformation than other platforms. What these findings demonstrate is that different kinds of authoritative information, public health messages, and platform disinformation responses are needed for the different categories of COVID-19 disinformation.

The fourth research question is whether there are differences in the categories for debunked COVID-19 claims of given veracity. We considered the following possible values of claim veracity: **False**—The given COVID-19 claim has been rated as false by the IFCN fact-checker who published the debunk; **Partially False**—the claim mixes true and false information, according to the fact-checkers; **Misleading**—the claim is rated as conveying misleading information; and **No evidence**—the fact-checkers found no evidence to prove the claim is true or not. The claim veracity information is extracted from the HTML tags on the IFCN debunk pages and is post-processed through string matching, as described in S2 Appendix. As shown in Table 6, 85% of the debunked disinformation in our dataset has been rated 'False' by the fact-checkers.

Fig 6 is a percentage stacked column chart of disinformation categories per claim veracity value. Overall, the distribution of topical categories per claim veracity value is no different from the overall category distribution in the entire dataset. The topical distribution of 'misleading' disinformation is slightly different from that of 'false' disinformation, as 'Community spread' has the largest proportion here. The 'No evidence' type distribution is clearly different as compared to the others, with 52.1% related to 'General medical advice', and 'Conspiracy Theories' as the second most mentioned category. This may be because for these two categories of disinformation it can be quite difficult to find solid scientific evidence that debunks them explicitly, especially in the earlier stages of the pandemic.

## 6 COVID-19 disinformation topics

In order to offer further insights into COVID-19 disinformation that spread between January and June 2020, we extracted the topics using CANTM by reusing the pre-trained M1 model (with labelled data), and only trained the M1 Classifier decoder and M2 model. Table 7 shows the examples of Class- Associated topics. Class- Associated topics are derived from $R_{ct}$ in M1

**Table 7. COVID-19 classification-associated topics from unlabelled data.**

| PubAuth | covid-19 president india china patients people ministry social police u.s. |
|---|---|
| CommSpread | people covid-19 died coronavirus false infected new outbreak photo shows |
| MedAdv | coronavirus water evidence prevent covid-19 experts health novel symptoms claims |
| PromActs | coronavirus claim says novel please article people outbreak trump donald |
| Consp | virus new evidence chinese created says novel video also predicted |
| VirTrans | spread claim health claims masks novel found china spreading facebook |
| VirOrgn | china outbreak covid-19 new market also novel indonesia shows claim |
| PubRec | video claim people shows novel outbreak lockdown times show old |
| Vacc | covid-19 vaccine novel claim testing disease said trump march new |

Classifier Decoder (Section 3.1.3) and the topics are directly associate with pre-defined classes, hence called Class- Associated topics.

Table 7 shows the top 10 topic words of the class-associated topics. As the topics are directly associated with the classifier prediction, the topic words are strongly linked with the pre-defined classes, and can be used as a global explanation of the classifier and for discovering concepts related to the classes. For example, the top topic words for Public Authority Action are 'president' and 'ministry'.

# 7 Related work

## 7.1 COVID-19 disinformation datasets and studies

Even though the COVID-19 infodemic is a very recent phenomenon, it has attracted very significant attention among researchers. Prior relevant COVID-19 'infodemic' research can be classified into one of two categories. The first one includes studies that are based entirely on information related to COVID-19 (without specifically distinguishing disinformation). The most relevant research in this category includes: the creation of a COVID-19 relevant Twitter dataset based on a time period covering the pandemic [24] or based on certain manually selected COVID-related hashtags [25–29]; sentiment analysis of information spread on Twitter [28, 30, 30, 36–41]; analysis of the spreading pattern of news with different credibility on Twitter [28, 31] and other social media platforms [32]; tweet misconception and stance dataset labelling and classification [42]; analysis of tweet topics using unsupervised topic modelling [30, 36–41, 43–49]; classification of informativeness of a tweet related to COVID-19 [50, 51]. Among these, the study most similar to ours is Gencoglu (2020) [52], which classifies tweets into 11 pre-defined classes using BERT and LaBSE [53]. However, the categories defined in [52] are generally different from ours, since ours are categories of disinformation specifically, whereas those of [52] aim to categorise all information relevant to COVID-19.

Our paper thus falls into the second category, which focuses specifically on research on COVID-19 disinformation. Related studies include: manually labelled likelihood of tweets containing false information and what types of damage could arise from this false information [34]; applying COVID-Twitter-BERT [54] to flag tweets for fact checking [55]; applying pre-trained NLP models including BERT to automatically detect false information [56–58]. As demonstrated in our experiments, the newly proposed CANTM model outperforms BERT-based models on this task.

Attention to the study of categories specific to COVID-19 disinformation is also found in previous research. Kouzy et. al. 2020 [33] study 673 tweets prior to February 27, 2020, and

report the proportion of the disinformation in different categories according to their manual labelling. Serrano et. al. 2020 [59] annotate 180 YouTube videos with two set of labels—a) disinformation or not; b) conspiracy theory or not—and propose several automatic classifiers using video comments based on pre-trained Transformer [60] models [61, 62] including BERT. Amongst these, the research closest to ours is Brennen et. al. (2020) [3], who carried out a qualitative study of the types, sources, and claims in 225 instances of disinformation across different platforms. In this paper, we adopted their disinformation categories; developed an automated machine learning method and a significantly larger annotated dataset; and extended the analysis on a much larger scale and over a longer time period.

## 7.2 Variational AutoEncoder (VAE) and supervised topic modelling

With respect to the computational methods, the following research is also relevant: **VAE based topic/document modelling** e.g. Mnih et. al. (2014) [63] trained a VAE based document model using the REINFORCE algorithm [64]; Miao et. al. [14] introduce Gaussian Softmax distribution, Gaussian Stick Breaking distribution and Recurrent Stick Breaking process for topic distribution construction. Srivastava et. al. in 2017 [65] proposed a ProdLDA that applies a Laplace approximation to re-parameterise Dirichlet distribution in VAE. Zhu et. al. [66] apply a Biterm Topic Model [67, 68] into the VAE framework for short text topic modelling. **Topic models with additional information** (e.g. author, label etc.): example work includes Supervised LDA [69], Labeled LDA [70], Sparse Additive Generative Model [71], Structural Topic Models [72], Author Topic Model [73], Time topic model [74] and topic model conditional on any arbitrary Features [15, 75]. **NVDM in text classification**: NVDM is also is apply NVDM as additional topic features [76, 77] in text classification. Compared with these approaches, CANTM is an asymmetric (different encoder input and decoder output) VAE that directly uses VAE latent variable as classification feature without external features, which enables the use of latent topics as classifier explanations. This explainability feature is highly beneficial for our specific use case.

## 8 Conclusion

This paper introduced the COVID-19 disinformation categories corpus, which provides manual annotation of debunked COVID-19 disinformation into 10 semantic categories. After quality control and a filtering process, the inter-annotator agreement average measured by Cohen's Kappa is 0.70. The paper also presented a new classification-aware topic model, that combines the BERT language model with the VAE document model framework, and demonstrates improved classification accuracy over a vanilla BERT model. In addition, the classification-aware topics provide class-related topics, which are: a) an efficient way to discover the class of (pre-defined) related topics; and b) a proxy explanation of classifier decisions.

The third contribution of this paper is a statistical analysis of COVID-19 disinformation which circulated between Jan and Jun 2020. It was conducted based on the automatically assigned category labels, and our main findings are:

1. The announcements from public authorities (e.g. WHO) highly correlate to public interest in COVID-19 and the volume of circulating disinformation. Moreover, disinformation about public authority actions is the dominating type of COVID-19 disinformation.

2. The relative frequency of the different disinformation categories varies throughout the different stages of the pandemic. Initially, the most popular category was 'Conspiracy theory', but then focus shifted to disinformation about 'Community spread' and 'Virus origin', only to shift again later towards disinformation about 'General medical advice'. As countries

began to take actions to combat the pandemic, disinformation about 'Public authority actions' began to dominate.

3. Different categories of disinformation are spread through different modalities. For instance, about half of the 'Virus origin' and 'Public reaction' disinformation posts are spread via video messages.

4. Facebook is the main originating platform of the disinformation debunked by IFCN fact-checkers, even though it has received much less attention than Twitter in related independent research.

## 9 Software and data

- COVID-19 disinformation category dataset: https://www.kaggle.com/dataset/fd97cd3b8f9b10c1600fd7bbb843a5c70d4c934ed83e74085c50b78d3db18443

- CANTM source code: https://github.com/GateNLP/CANTM

- Webservice: https://cloud.gate.ac.uk/shopfront/displayItem/covid19-misinfo

- REST API: https://cloud-api.gate.ac.uk/process-document/covid19-misinfo

## Supporting information

**S1 Appendix. Data structure and example IFCN web page.**
(PDF)

**S2 Appendix. The string matching process.**
(PDF)

**S3 Appendix. Rule-based extraction of media type.**
(PDF)

**S4 Appendix. Definitions of the COVID-19 disinformation categories.**
(PDF)

**S5 Appendix. Deriving the ELBO.**
(PDF)

**S6 Appendix. Extra experimental details.**
(PDF)

**S7 Appendix. CANTM confusion matrix.**
(PDF)

**S8 Appendix. Classification-aware topics examples.**
(PDF)

**S1 Data.**
(ZIP)

**S2 Data.**
(ZIP)

## Author Contributions

**Conceptualization:** Xingyi Song.

**Data curation:** Xingyi Song, Ye Jiang, Iknoor Singh, Diana Maynard.

**Formal analysis:** Xingyi Song.

**Funding acquisition:** Kalina Bontcheva.

**Methodology:** Xingyi Song.

**Project administration:** Diana Maynard, Kalina Bontcheva.

**Resources:** Iknoor Singh.

**Software:** Xingyi Song, Johann Petrak.

**Supervision:** Diana Maynard, Kalina Bontcheva.

**Writing – original draft:** Xingyi Song.

**Writing – review & editing:** Johann Petrak, Ye Jiang, Iknoor Singh, Diana Maynard, Kalina Bontcheva.

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
