## [Decision Letter · Decision Letter 0]

4 Dec 2020

PONE-D-20-33400

Classification Aware Neural Topic Model for COVID-19 Disinformation Categorisation

PLOS ONE

Dear Dr. Song,

Thank you for submitting your manuscript to PLOS ONE. After careful consideration, we feel that it has merit but does not fully meet PLOS ONE’s publication criteria as it currently stands. Therefore, we invite you to submit a revised version of the manuscript that addresses the points raised during the review process.

We look forward to receiving your revised manuscript.

Kind regards,

Sanda Martinčić-Ipšić, PhD

Academic Editor

PLOS ONE

Journal Requirements:

2.In your Data Availability statement, you have not specified where the minimal data set underlying the results described in your manuscript can be found. PLOS defines a study's minimal data set as the underlying data used to reach the conclusions drawn in the manuscript and any additional data required to replicate the reported study findings in their entirety. All PLOS journals require that the minimal data set be made fully available. For more information about our data policy, please see http://journals.plos.org/plosone/s/data-availability.

3.Thank you for submitting the above manuscript to PLOS ONE. During our internal evaluation of the manuscript, we found significant text overlap between your submission and the following previously published work, of which you are an author: http://eprints.whiterose.ac.uk/164746/

Please revise the manuscript to rephrase the duplicated text, cite your sources, and provide details as to how the current manuscript advances on previous work. Please note that further consideration is dependent on the submission of a manuscript that addresses these concerns about the overlap in text with published work.

Reviewers' comments:

Reviewer's Responses to Questions

**Comments to the Author**

1. Is the manuscript technically sound, and do the data support the conclusions?

Reviewer #1: Yes

Reviewer #2: Yes

2. Has the statistical analysis been performed appropriately and rigorously? 

Reviewer #1: N/A

Reviewer #2: Yes

3. Have the authors made all data underlying the findings in their manuscript fully available?

Reviewer #1: Yes

Reviewer #2: No

4. Is the manuscript presented in an intelligible fashion and written in standard English?

Reviewer #1: Yes

Reviewer #2: No

5. Review Comments to the Author

Reviewer #1: Authors have compiled a COVID-19 related dataset and labelled it with 10 categories. They manually annotated a part of data with a good enough Kappa score. To automatically categorize data they used BERT model with a combination of encoder-decoder networks. Also the input for BERT is word-piece-based (as standard for BERT dictionary) while for E-D nework is simple BOW representation. At the end, authors compare and evaluate their results.

The paper is well structured, well written. Most of the work is justified and also data and DEMO version of the deployed algorithm is available (full after the publication) online.

I have no major comments but I propose authors to update the following parts:

- In the beginning of Chapter 4 authors should define the D_{KL}.

- In the Figure 1 authors can ommit some basic explanations (e.g. linear layer) as it is supposed the reader already understands BERT model which i more complex. Also in the figure not the same words are used.

- In the results I am not sure what is NVDMb and NVDMo? Also, as accuracy is reported, the percentage of majority class should be mentioned (otherwise reader needs to calculate it from Table 4?).

- Authors will publish annotated data. Could authors also publish automatically annotated data by their algorithm or publish their code. That would be useful for reproducibility and further comparitons.

- Authors sometimes start sentence with a formula (e.g. "p(x|z) is the generation"....) or reference (e.g. "[8] introduce a"...). I propose to reformat sentences in a way that they do not start like these.

Reviewer #2: In the manuscript "Classification Aware Neural Topic Model for COVID-19 Disinformation Categorisation" authors perform topic modelling of COVID-19 disinformation using neural networks. They combine BERT model with Variational Autoencoder and define a CANTM model for a topic generation. The proposed model is evaluated in terms of standard evaluation measures (accuracy, macro F- score and perplexity). The reported results show that the proposed model outperforms some other state-of-the-art approaches and human annotators. The evaluation procedure seems to be correctly implemented. However, the whole manuscript is too extensively written and certain parts of the described approach need to be clarified by explaining the experiment in a more concise text.

In general, this research is interesting and valuable, although the rest of the manuscript is not easy to follow. Overall, the manuscript has certain shortcomings, which need to be improved before the work is good enough to be recommended for publication.

My suggestions and comments are as follows.

1. Abstract should be rewritten. Now it seems to be slightly misleading because it is written that this research will develop “computational methods to support research on COVID-19 disinformation debunking and its social impact”. In the abstract, it should be emphasized that the main focus of their research is to identify the topic of fake news, not to identify fake news. Furthermore, this abstract is missing an overview of research method and insight into the results.

2. In the introductory section authors describe the motivation, main goals and challenges of their research- The scientific contributions are clearly stated as well. My suggestion is to add one paragraph with concise descriptions of all experiments.

3. Section Dataset Structure is written with too many details. The first part of the Section related to Table 1, together with this table can be moved to the Supplementary materials and leaving only data about dataset statistics.

4. Furthermore, this second Section about the data structure can be a subsection of the section which describes the experiment.

5. The third Section about disinformation category labelling is also too extensive.

6. Section about related work needs to be extended with more references that are relevant for this research. I suggest the authors to include more publications that use BERT model in similar NLP tasks.

6. PLOS authors have the option to publish the peer review history of their article (what does this mean?). If published, this will include your full peer review and any attached files.

Reviewer #1: **Yes: **Slavko Žitnik

Reviewer #2: No

---

## [Author Response · Author response to Decision Letter 0]

18 Jan 2021

Dear Reviewers and AE,

We would like to express our appreciation for your insightful comments, constructive suggestions and valuable time for the paper. We sincerely appreciate the encouragement in your comments. We are happy to report that we have addressed the issues that the reviewers raised to the best of our capacity. We have revised the structure and addressed the editorial problems addressed from the reviewers’ suggestions. 

A color marked-up copy of our manuscript that highlights changes made to the original version is included. Red marks indicate major changes/revisions of paragraphs. Orange marks indicate minor grammatical/structure changes. Blue marks indicate new added paragraphs.

The responses to each comment in detail are as follows.

Response Reviewer #1:

1. In the beginning of Chapter 4 authors should define the D_{KL}.

We added the definition of D_{KL} in Line 141 as “Kullback–Leibler divergence”

2. In the Figure 1 authors can ommit some basic explanations (e.g. linear layer) as it is supposed the reader already understands BERT model which i more complex. Also in the figure not the same words are used.

We added the definition of linear layer in Equation 2 and Line 190. We also updated the figure and equations to have consistent wording.

3. In the results I am not sure what is NVDMb and NVDMo? Also, as accuracy is reported, the percentage of majority class should be mentioned (otherwise reader needs to calculate it from Table 4?).

We have moved the details of experiment settings from supplemental material to the main content. The definition of NVDMb and NVDMo now in the Line 234- 236, which is “1) original NVDM as described in [8] (“NVDMo” in the results ); 2) NVDM with BERT representation (“NVDMb” in the results).”

4. Authors will publish annotated data. Could authors also publish automatically annotated data by their algorithm or publish their code. That would be useful for reproducibility and further comparitons.

We will publish the source code and the data used in this paper. Links to the code and data are added in Section 9 Software and Data

5. Authors sometimes start sentence with a formula (e.g. "p(x|z) is the generation"....) or reference (e.g. "[8] introduce a"...). I propose to reformat sentences in a way that they do not start like these.

Thanks for the suggestion, we have revised the manuscript accordingly 

Response Reviewer #2:

1. Abstract should be rewritten. Now it seems to be slightly misleading because it is written that this research will develop “computational methods to support research on COVID-19 disinformation debunking and its social impact”. In the abstract, it should be emphasized that the main focus of their research is to identify the topic of fake news, not to identify fake news. Furthermore, this abstract is missing an overview of research method and insight into the results.

Thanks for your suggestion, we have rewritten the abstract and parts of the introduction based on your suggestion. 

2. In the introductory section authors describe the motivation, main goals and challenges of their research- The scientific contributions are clearly stated as well. My suggestion is to add one paragraph with concise descriptions of all experiments.

A concise description of experiments is added in Line 59-64

3. Section Dataset Structure is written with too many details. The first part of the Section related to Table 1, together with this table can be moved to the Supplementary materials and leaving only data about dataset statistics.

Thanks for your suggestion, we have now significantly reduced the content of Section Dataset Structure. We left Table 1 (with reduced rows) in the main content, because dataset building is one of the main contributions of this work, and the table is a clear way to present the structure of the data. 

4. Furthermore, this second Section about the data structure can be a subsection of the section which describes the experiment.

Thanks for your suggestion, we now merged this section with Section 5 “COVID-19 Disinformation Analysis and Discussion” 

5. The third Section about disinformation category labelling is also too extensive.

Thanks for your suggestion. We have revised the context and merged this section with Section 2 and renamed the sections as “Dataset and Annotation”. Data annotation and the cleaning process are one of the most important steps in our work. This ensures the correctness of the category labeling work and affects the correctness of our experiment and analysis process described in the later sections. We described this process within one page in the current version of the manuscript.

6. Section about related work needs to be extended with more references that are relevant for this research. I suggest the authors to include more publications that use BERT model in similar NLP tasks.

Thanks for the suggestion, we now enriched the related work section, and highlighted some of the work using BERT model.

Response editor Sanda Martinčić-Ipšić’s queries:

We are using the PLOS ONE official Latex template and ensured the manuscript meets PLOS ONE's style requirements 

The full dataset is publicly available at:

www.kaggle.com/dataset/fd97cd3b8f9b10c1600fd7bbb843a5c70d4c934ed83e74085c50b78d3db18443

The source code is publicly available at:

https://github.com/GateNLP/CANTM

We added one more section “Section 9 Software and data” in the updated manuscript reporting the availability of source code and dataset.

3. During our internal evaluation of the manuscript, we found significant text overlap between your submission and the following previously published work, of which you are an author: http://eprints.whiterose.ac.uk/164746/

The work http://eprints.whiterose.ac.uk/164746/ is a copy of our arxiv version of the paper (please note the link to the paper is arxiv). White Rose Research Online is a repository that automatically collects research outputs from University of Sheffield and two other universities. The collection includes an arxiv preprint. 

We deeply appreciate your time reviewing our manuscript,

Thank you and best regards.

Xingyi Song

---

## [Editor Report · Decision Letter 1]

2 Feb 2021

Classification Aware Neural Topic Model for COVID-19 Disinformation Categorisation

PONE-D-20-33400R1

Dear Dr. Song,

We’re pleased to inform you that your manuscript has been judged scientifically suitable for publication and will be formally accepted for publication once it meets all outstanding technical requirements.

Kind regards,

Sanda Martinčić-Ipšić, PhD

Academic Editor

PLOS ONE

Additional Editor Comments (optional):

This manuscript relates to the ongoing outbreak of coronavirus. Given this, I checked the revision, your response to reviewers and data and SW availability. I am glad that the current manuscript revision has addressed all issues adequately and meets required PlosONE criteria. 
---

## [Editor Report · Acceptance letter]

4 Feb 2021

PONE-D-20-33400R1 

Classification Aware Neural Topic Model for COVID-19 Disinformation Categorisation 

Dear Dr. Song:

I'm pleased to inform you that your manuscript has been deemed suitable for publication in PLOS ONE. Congratulations! Your manuscript is now with our production department. 

Kind regards, 

on behalf of

Dr. Sanda Martinčić-Ipšić 

Academic Editor

PLOS ONE